# The Updating of Biological Functions of Methyltransferase SETDB1 and Its Relevance in Lung Cancer and Mesothelioma

**DOI:** 10.3390/ijms22147416

**Published:** 2021-07-10

**Authors:** Li Yuan, Boshu Sun, Liangliang Xu, Limin Chen, Wenbin Ou

**Affiliations:** Zhejiang Provincial Key Laboratory of Silkworm Bioreactor and Biomedicine, Department of Biopharmaceuticals, College of Life Sciences and Medicine, Zhejiang Sci-Tech University, Hangzhou 310018, China; 2019339902100@mails.zstu.edu.cn (L.Y.); 2019339902095@mails.zstu.edu.cn (B.S.); 201920201073@mails.zstu.edu.cn (L.X.); 2018339901294@mails.zstu.edu.cn (L.C.)

**Keywords:** methyltransferase, SETDB1, function, lung cancer, mesothelioma

## Abstract

SET domain bifurcated 1 (SETDB1) is a histone H3 lysine 9 (H3K9) methyltransferase that exerts important effects on epigenetic gene regulation. SETDB1 complexes (SETDB1-KRAB-KAP1, SETDB1-DNMT3A, SETDB1-PML, SETDB1-ATF7IP-MBD1) play crucial roles in the processes of histone methylation, transcriptional suppression and chromatin remodelling. Therefore, aberrant trimethylation at H3K9 due to amplification, mutation or deletion of SETDB1 may lead to transcriptional repression of various tumour-suppressing genes and other related genes in cancer cells. Lung cancer is the most common type of cancer worldwide in which SETDB1 amplification and H3K9 hypermethylation have been indicated as potential tumourigenesis markers. In contrast, frequent inactivation mutations of SETDB1 have been revealed in mesothelioma, an asbestos-associated, locally aggressive, highly lethal, and notoriously chemotherapy-resistant cancer. Above all, the different statuses of SETDB1 indicate that it may have different biological functions and be a potential diagnostic biomarker and therapeutic target in lung cancer and mesothelioma.

## 1. Introduction

Epigenetic gene regulation eventually leads to changes in gene function and phenotype without changes in DNA sequence [1]. Among epigenetic gene regulation mechanisms, histone methylation is extremely important and participates in gene expression and chromatin organization [2]. The process of histone methylation involves two enzyme systems, histone methyltransferases (HMTs) and histone demethylases (HDMs), the imbalance of which is closely related to tumourigenesis [3].

Histone methylation occurs at the arginine and lysine residues of histone 3 (H3) and histone 4 (H4). Arginine and lysine residues can be monomethylated or dimethylated, and lysine residues can also be trimethylated [4,5]. Some common histone methyltransferases and their related cancers are shown in Table 1. Histone lysine methyltransferases (HKMTs) are important epigenetic modificatory enzymes containing SET domains. Being responsible for the enzymatic activity, the SET domain is included in most HKMTs. The HKMTs involved in human tumourigenesis consist mainly of the multicomb complex SMYD, NSD, SETD, SUV39, EHMT, MLL, DOT1L, and PRC2 [6,7]. SET domain bifurcated 1 (SETDB1), belonging to the SUV family of H3K9 methyltransferases, exploits its H3K9me1/2/3 functions by the conserved SET domain [8]. H3K9 trimethylation possesses the ability to form heterochromatin, which silences gene transcription and remodels the chromatin structure [1,9,10]. Overexpression and downregulation of SETDB1 have been widely found in various cancers, but the detailed mechanisms are still unclear [2,11,12,13,14,15,16,17,18,19,20,21,22]. Here, we summarize the biological functions of SETDB1 and its therapeutic relevance in lung cancer and mesothelioma, describing possible diagnostic and therapeutic prospects.

## 2. Biological Functions of SETDB1

SETDB1 is primarily involved in the processes of histone methylation, transcriptional suppression and chromatin gene silencing as well as chromatin remodelling in cells [9], either to maintain the structure of chromatin or to control the expression of specific genes. Simultaneously, SETDB1 exerts many other physiological functions, including cell apoptosis [30,31], antiviral response [32,33]; the establishment, growth and proliferation of pluripotent embryonic stem cells [34]; proviral silencing during embryogenesis and postnatal development [35]; and X-chromosome inactivation [36]. Most of these functions are dependent on its methyltransferase activity. More importantly, the expression of SETDB1 is upregulated in a variety of tumours, indicating that SETDB1 is closely related to the occurrence and development of cancers [21].

As a histone lysine methyltransferase, aberrant SETDB1 expression is frequently found in lung cancer and malignant mesothelioma [2,12,13,16,17,18,37,38]. Overexpression in lung cancer and inactivation in mesothelioma indicate disparate functions of SETDB1 in different types of cancers. To further understand the biological roles and mechanisms of SETDB1, we summarized the functions of SETDB1 as well as its therapeutic relevance in lung cancer and mesothelioma. SETDB1 may be a clinical diagnostic biomarker and therapeutic target in the future.

### 2.1. The Methylation of Histone H3 Lysine 9 by SETDB1

Histone methylation is a major regulator of epigenetic modification and plays a key regulatory role in gene expression. Histone lysine methylation is an important participant among these regulators and is associated with malignant conversion of tumours and the regulation of various physiological activities [39]. As a H3K9 methylase, SETDB1 was found to specifically methylate the lysine residue of histone 3 at site 9 [9], through which it maintains the structure of DNA and controls gene expression by regulating the extent of DNA compaction [40]. During the process of methylation, the SET domain of SETDB1 utilizes S-adenosylmethionine (SAM) to methylate the ε-amino group of the lysine residue [6,41].

SETDB1 individually dimethylates H3K9. The interaction with the human homologue of murine ATFa-associated modulator (hAM) (the cofactor of SETDB1) enhances its enzyme activity, which facilitates the SETDB1-dependent conversion at H3 in euchromatic regions from dimethyl H3K9 to a trimethyl state [10]. The trimethylation of H3K9 is associated with gene suppression, while monomethylation is associated with gene excitation [42]. Thus, association with hAM increases SETDB1-dependent transcriptional repression on a chromatin template [10].

### 2.2. Gene Transcription Silencing by the SETDB1-KRAB-KAP1 Complex

The methyltransferase activity of SETDB1 at the H3K9 position in the nuclear euchromatic locus facilitates the combination of the chromo-domain of heterochromatin protein 1α (HP1α) and methylated Lys, which promotes HP1 deposition [9]. HP1 might dimerize with methylated H3K9 in close vicinity and form compacted heterochromatin [43], which contributes to gene silencing. KRAB-associated protein-1 (KAP1), a transcriptional intermediary factor acting as a molecular scaffold among many transcriptional regulatory complexes, was reported to be able to associate with both SETDB1 and HP1α [9]. The connection between SETDB1 and HP1 has been previously confirmed [44]. Specifically, the repression domain of KAP1 binds to the chromo shadow domain of HP1α [45,46]. In addition, KAP1, a common corepressor for the KRAB zinc finger protein (KRAB-ZFP) family (the largest family of sequence-specific DNA binding repressors [47]) of transcriptional repressors [48], connects to KRAB-ZFP through its 75-amino acid KRAB box to form a KRAB-KAP1 repression complex that establishes local microenvironments of heterochromatin at the N-terminal tail of histone H3. As a consequence, the heterochromatin status suppresses gene transcription at specific sites [9,48,49]. Thus, the SETDB1-KRAB-KAP1 repression complex leads to enriched SETDB1, H3K9me3, KRAB, KAP-1, and HP1 at the promoter sequences of specific euchromatic genes, which induces conversion of the target genes from euchromatin to facultative heterochromatin and subsequently represses gene expression [9,50,51,52].

In mammalian genomes, endogenous retroviruses (ERVs) comprise approximately 8% of the human genome [52]. Although retrotransposition contributes to genome diversification evolution and adaption, it can also lead to genome instability, insertional mutagenesis, or transcriptional perturbation, which is often harmful to host cells [53,54,55,56]. ERVs are responsible for 10–12% of the total spontaneous mutations in mice [57]. Aberrant expression of ERVs could alter the expression of neighbouring genes, some of which act as proto-oncogenes to transform host cells [58]. Therefore, multiple defence mechanisms have evolved to maintain the integrity of the genome and transcriptome against retroelement transposition, among which H3K9 methylation modification is important [52] and especially mediated by SETDB1 [32,33,59,60].

ERV silencing is initiated by the recruitment of KAP1 to the target by members of the KRAB-ZFP family. Then, the KAP1/KRAB-ZFP complex is thought to be a pivotal factor for SETDB1 recruitment to retroelements [55]. Simultaneously, HP1 and the NuRD complex are recruited by KAP1 [49]. In detail, KAP1 was discovered to autosumoylate its bromodomain to recruit both the NuRD complex and SETDB1 to the promoter regions of genes modulated by KRAB-ZFPs, which could establish a silent chromatin state by H3K9me3 at genes targeted by KAP1 [9,35]. Recent studies have reported that mouse embryonic stem cells (mESCs) enhance recruitment of SETDB1 to ERV retrotransposition and formation of a KAP1 suppression complex to repress proviral molecules [61]. Furthermore, Peter J. Thompson identified that RNA-binding protein and transcription cofactor heterogeneous nuclear ribonucleoprotein K (hnRNP K) are necessary for the SETDB1-dependent proviral silencing process, which acts as a binding partner of the SETDB1-KAP1 complex by direct interaction in mESCs [61]. Recently, studies of SETDB1 knockout adult mice and differentiated cells demonstrated that SETDB1 also represses retroelements in somatic cells, such as B lymphocytes [62], T lymphocytes [63], neural progenitor cells (NPCs) [64], and immortalized mouse embryonic fibroblasts (iMEFs) [65]. Mechanistically, SETDB1 is recruited to its target regions by KRAB-ZNF/KAP1 to take effect [66,67], confirming a more general role of SETDB1 in suppressing retroelements.

Recently, SETDB1-dependent ERV repression in cancer cells was also confirmed to be associated with evading recognition by the immune system [68,69,70]. As a negative regulator of innate immunity, SETDB1 was reported to repress the expression of ERVs through its H3K9-methylating function to preclude the immune response of B cells induced by ERVs, which enables acute myeloid leukaemia (AML) cells to escape innate immunity [68,69]. Above all, the model of SETDB1-KRAB-KAP1 initiated by the recruitment of KAP1 plays a pivotal role in maintaining genome stability and immune regulation, especially through ERVs silencing [70,71].

### 2.3. Gene Transcription Silencing by Interaction between SETDB1 and DNMT3A

A direct interaction between the N-terminal domain of SETDB1 and the plant homeodomain (PHD) of DNA methyltransferase 3A (DNMT3A) in vivo and in vitro has been shown to facilitate gene transcriptional repression [1]. The repression of the tumour-suppressing gene *RASSF1A* by a high frequency of methylation at its promoter is widely found in lung, breast, pancreas, kidney, liver, and other cancers [72]. The SETDB1-DNMT3A complex was essential for repressing the promoter of the *p53BP2* gene in HeLacells and the *RASSF1A* gene in MDA-MB-231 breast cancer cells [1,72]. The study also confirmed that SETDB1 is recruited by methyl-CpG DNA Binding Domain Protein 1 (MBD1) to the promoter region of *p53BP2* (it seems that MBD1 does not mediate the recruitment of SETDB1 to *RASSF1A*), and the co-occupation of DNMT3A and SETDB1 leads to a hypermethylated gene promoter [1]. Further study found that SETDB1, DNMT3A, and histone deacetylase 1 (HDAC1) formed a repressive functional complex. During the process, the SETDB1-HDAC1 complex is recruited to the promoter at first to establish trimethylated H3K9 status and represses gene transcription. DNMT3A is later recruited to form the SETDB1-HDAC1-DNMT3A complex and enhance DNA methylation to permanently inactivate the gene. It is a self-propagating epigenetic cycle, in which SETDB1 methylates H3K9 and recruits DNMT3A to reinforce and maintain DNA methylation [1]. Therefore, MBD1-dependent SETDB1-HDAC1 recruitment and SETDB1-mediated H3-K9 methylation initiate the establishment of trimethylated H3K9 status at *p53BP2* promoter. SETDB1 recruits DNMT3A which reinforces DNA methylation to connect histone methylation and DNA methylation.

### 2.4. Gene Transcription Silencing by Interaction between SETDB1 and PML

Promyelocytic leukaemia nuclear bodies (PML-NBs) are large proteinaceous structures that participate in diverse cellular processes such as apoptosis, transcription, cellular senescence, neoangiogenesis, DNA damage response, antiviral response and maintenance of genomic stability, by interacting with multiple proteins [30,31]. Sunwha Cho et al. demonstrated a combination of endogenous SETDB1 and PML proteins from the cleavage stages of development in mice, and their colocalization at PML-NBs was also confirmed. SETDB1 was found to harbor dual functions in this complex. On the one hand, as a necessary part to maintain the structural integrity of PML-NB, SETDB1 physically interacts with the PML protein with its SIM motif [73,74]. On the other hand, SETDB1 acts as a transcriptional regulator of PML-NB-associated genes. The researchers discovered that SETDB1 occupies the promoter of *inhibitor of DNA binding 2* (*ID2*) and methylates H3K9 to prevent *ID2* binding to RNA polymerase II, through which it suppresses ID2 expression. This inhibitory progress also depends on the integrity of the PML-NB structure [73,74]. SETDB1-mediated local heterochromatin formation involves a self-reinforcing mechanism: SETDB1-produced H3K9me3 marks recruit HP1 to localize at PML-NB foci [75]. With the aid of HP1, SETDB1 forms a solid platform of heterochromatin to which PML-NB can be riveted [74]. Thus, SETDB1 not only maintains the structure of PML-NB, but also represses the expression of PML-NB-associated genes such as *ID2* by trimethylating specific gene locus.

### 2.5. X-Chromosome Inactivation by SETDB1-ATF7IP-MBD1 Complex

X-Chromosome inactivation (XCI), an epigenetic silence caused by heterochromatin formation during the early phase of female mammalian embryonic development, is maintained throughout the lifetime of somatic cells [36]. According to many studies, the accumulation of lncRNA Xist on the X chromatin is the master factor leading to the initiation and maintenance of XCI [76,77,78]. Interestingly, SETDB1 was also discovered to be related to XCI [36,79]. SETDB1 is the most required methyltransferase to silence approximately 150 genes, which facilitates gene silencing and maintains XCI by alternating the conformation of the whole inactive X chromosome [79]. Minkovsky et al. suggested that during XCI, activating transcription factor 7 interacting protein (ATF7IP or MCAF1), as a bridging factor, interacts with both SETDB1 and the transcriptional repressor domain of MBD1 to form the transcriptional repressor complex SETDB1-ATF7IP-MBD1, which leads to histone H3K9 trimethylation on the inactive X chromosome (Xi) [36,80]. Additional studies confirmed that the formation of the MBD1-chromatin assembly factor-1 (CAF-1) chaperone complex initiates the formation of the transcriptional repressive complex by mediating SETDB1 recruitment to the large subunit of CAF-1 [74,81,82] and maintaining XCI in somatic cells [36,83]. H3K9me3, considered a marker of heterochromatin [73], was enriched at the intergenic, poor gene and repetitive regions of Xi. The heterochromatin structure is inherited during DNA replication through association with MBD1 and ATF7IP [81].

Recently, the location of SETDB1 inside the nucleus was discovered to be mediated by ATF7IP, which increases the ubiquitination of SETDB1 to promote its enzymatic activity [84,85,86]. The fibronectin type-III (FNIII) domain of ATF7IP plays a certain role in the transcriptional repression function mediated by the SETDB1-ATF7IP complex. However, the FNIII domain seems to be unrelated to the nuclear localization of SETDB1 and to the ATF7IP-dependent integrated retroviral transgenes silencing [86]. Above all, by alternating the conformation of the X chromosome through SETDB1-ATF7IP-MBD1 complex, SETDB1 also plays an important role in XCI.

### 2.6. Remodelling of Chromatin Associated with SETDB1 Expression

In protein posttranscriptional modification, histone methyltransferases regulate the folding and remodelling of chromatin by modifying the N-tail of histones [39]. SETDB1 is a histone H3K9-specific methyltransferase that associates with various transcription factors to regulate gene expression via chromatin remodelling, reflecting that SETDB1 is closely related to chromatin modification and heterochromatin formation [18,87,88]. Recently, some studies reported that the master silencing factor KAP1 recruits other factors to establish interstitial heterochromatin in mESCs, especially SETDB1 [52,88]. During the formation of the SETDB1-KRAB-KAP1 repressive complex mentioned above, the interaction between HP1α and KAP1 triggers the conversion of target foci from euchromatin to heterochromatin, thereby inhibiting gene expression [9,48]. H3K9 methyltransferases have also been confirmed to be the “gatekeepers” of chromatin condensation. Methylated H3K9 can recruit H1 linker histones and HP1, while the localization of histone H1 is associated with the gliding of nucleosomes [89]. In addition, SETDB1 can also combine with DNMTs or HDACs to remodel the chromatin structure [90]. Many studies have confirmed that SETDB1 plays an essential role in chromatin remodelling, especially heterochromatin formation, through interacting with other factors and its H3K9 methylation function.

### 2.7. Early Embryo Development Associated with SETDB1

In mouse embryos, SETDB1 holds a prominent position in early development. SETDB1 dysfunction has been found to induce the earliest lethality around peri-implantation compared with other H3K9-specific HKMTs such as SUV39h1 (nonessential) [91], G9a (die around day 9.5) [92], and GLP (die around day 9.5) [93]. Cho et al. found that SETDB1 discontinuously appears in the pronucleus after fertilization, and its expression in males is higher than its expression in females [94,95], indicating that SETDB1 may participate in the restructuring of sperm-derived chromatin [20,96]. The expression of SETDB1 turns to a diffuse pattern until the 2-cell stage [96] but temporarily fades in the 8-cell stage before reappearing in a spotted form again in the blastocyst [73,97]. During the blastocyst phase, restored SETDB1 localizes to PML-NB foci [97] and is expressed equally in the inner cell mass (ICM) and trophectoderm (TE) cells. SETDB1 is then expressed only in the ICM during the blastocyst outgrowth phase [96]. The varying expression patterns of SETDB1 during the preimplantation stage suggest crucial functions of SETDB1 in mouse early embryonic development. The catalytic activity of SETDB1 is necessary in meiosis and early oogenesis, without it, a decrease in the number of mature eggs as a result [98,99].

### 2.8. Embryonic Stem Cell Development Associated with SETDB1

Blastocysts comprise ICM and TE cells. The punctate pattern of SETDB1, as mentioned in 2.7., was only expressed in ICM-derived Oct4-positive cells during the blastocyst outgrowth phase. Furthermore, ESCs can be established in vitro by an extended culture of ICM cells [74], which is consistent with the result that SETDB1-null blastocysts fail to give rise to ESCs in vitro [100], suggesting the pivotal role of SETDB1 in the establishment and/or maintenance of ESCs.

SETDB1 was found to maintain the pluripotency of mESCs by repressing differentiation-associated genes [74] and trophoblastic genes [101]. Conditional knockout of SETDB1 in mESCs induced the expression of trophectoderm differentiation markers such as caudal type homeobox 2 (Cdx2), transcription factor AP-2 alpha (Tcfap2a), and heart and neural crest derivatives expressed 1 (Hand1) [102]. Additional data found that SETDB1 interacts with Oct4, which in turn recruits SETDB1 to silence trophoblast-associated genes. These findings demonstrate that SETDB1 restricts the extraembryonic trophoblast lineage potential of pluripotent cells [102]. Recently, a study demonstrated that SETDB1 restricts the transition from pluripotency to totipotency in ESCs by silencing ERV and relevant genes [103]. SETDB1 is also essential to mESC self-renewal [101,104,105]. Lack of SETDB1 resulted in downregulation of *Nanog*, *SRY-box 2* (*Sox2*) and *POU class 5 homeobox 1* (*Pou5f1*) genes associated with induced pluripotent stem cell (iPSC) reprogramming and positive regulation of pluripotency and self-renewal of mESCs [102].

A recent study demonstrated that the H3K9 methylation reader M-phase phosphoprotein 8 (MPP8) cooperates with SETDB1 physically and functionally to coregulate a great number of common genomic targets, especially the DNA satellite, which silences satellite DNA repeats in mESCs [80].

### 2.9. Brief Summary of the Functions of SETDB1

Herein, we summarize the SETDB1 function as shown in Figure 1. As mentioned above, SETDB1 primarily represses gene expression with chromatin remodelling based on its methyltransferase activity at H3K9, through which it exerts various functions, including ERVs silencing [68,69], XCI [36,79] and tumour-suppressive genes repression [1,72]. When SAM is available, SETDB1 could deliver its H3K9 trimethylation function with the help of hAM [6,10,41]. During the process of ERVs silencing, KRAB-ZFP initiates the formation of the transcriptional repressive complex by recruiting KAP1 to the target [55]. With the recruitment of other factors and construction of H3K9 trimethylation, it induces heterochromatin formation and ERVs silencing [9,35]. During the process of p53BP2 repression, MBD1 mediates the recruitment of SETDB1 while SETDB1 facilitates H3K9 trimethylation and the recruitment of DNMT3A, which associates histone methylation and enhanced DNA methylation [1]. SETDB1 can also associate with PML-NB complex and repress the expression of PML-NB-associated genes with cooperation of HP1 [73,74]. In the context of XCI, MBD1 initiates the recruitment of SETDB1 to CAF-1 [74,81,82], and the formation of SETDB1-ATF7IP-MBD1 promotes XCI [36,80] (Figure 1A). Moreover, SETDB1 is essential to early embryo development, whose expression patterns vary during the preimplantation stage [73,96,97] (Figure 1B). In blastocyst outgrowth phase, the interaction between SETDB1 and Oct4 in mESCs recruits SETDB1 to silence differentiation-associated and trophoblast-associated genes, through which SETDB1 maintains the pluripotency of mESCs [74,101,102]. Lack of SETDB1 in mESCs was found to induce the expression of trophectoderm differentiation markers [102] and downregulate the genes associated with iPSC reprogramming and self-renewal [101,104,106] (Figure 1B).

## 3. Tumourigenesis Associated with Expression of SETDB1

Recent studies have shown that the dysregulation of epigenetic inheritance is a vital contributor to cancer formation [105]. As an important participator of epigenetics, aberrant methylation at H3K9 induced by the histone methyltransferase SETDB1 may contribute to tumourigenesis in breast cancer, colorectal cancer, glioma, ovarian cancer, lung cancer and melanoma [13,20,21,107].

Under various carcinogenic conditions or regulation of related genes, the transcription of SETDB1 was reported to be prominently elevated [108]. Amplified or overexpressed SETDB1 levels often induce excessive H3K9 trimethylation at promoter regions, which changes the local structural dynamics of chromatin and finally leads to abnormally silenced tumour-suppressive genes and tumourigenesis [107]. For example, the expression of *Tp53* in pancreatic ductal adenocarcinomas [109,110] and colorectal cancer [14,21,110,111], *p16-INK4A* in melanomas [112], *p53BP2* in hela cell line of cervical cancer [1], *APOE*, and *HoxA* in lung cancer [21], as well as *RASSF1A* in breast cancer cells [72], are all repressed by SETDB1. Therefore, SETDB1 has been confirmed as an oncogene in human melanomas and lung cancers [2,113,114]. However, downregulation of SETDB1 was also found in metastatic lung cancer [12,13,115,116,117], less aggressive osteosarcomas [21] and AML [68,69,74,118,119,120], indicating a tumour suppressive role of SETDB1 in different cancer types and stages. Moreover, inactivation mutation of SETDB1 was found in malignant pleural mesothelioma (MPM), suggesting that it may lose function and act as a tumour suppressor in mesothelioma [16,17]. The following are the updated tumourigenesis associations with SETDB1 expression in lung cancer and mesothelioma.

### 3.1. Lung Cancer

Lung cancer can be categorized into two major clinicopathological types: small-cell lung carcinoma (SCLC) and nonSCLC (NSCLC). NSCLC accounts for 85% of all lung cancers and is further classified into adenocarcinoma (ADC), squamous cell carcinoma (SCC), and large-cell carcinoma (LCC). NSCLC, especially ADC and SCC, has relatively higher mutation rates than the mean rate of other lung cancer subtypes [121].

According to the latest statistical data, the lung cancer death rate has fallen continuously from its peak from 1991 to 2018 in the United States of America (USA), attributed to improved treatment in early detection and treatment as well as decades of comprehensive tobacco control programs since the 1990s [122]. However, lung cancer remains the leading cause of cancer deaths [123] and will still be the top cancer type for the estimated new cancer deaths in both males and females [122].

In lung cancer, hyperactivation of SETDB1 was found to influence various pathways, such as the WNT, MAPK, Toll-like receptors (TLRs), focal adhesion, and JAK-STAT pathways [11]. Some reports have also focused on the possible therapeutic target of SETDB1 in lung cancer treatment [38,124,125]. For example, SETDB1 positively stimulated the WNT-β-catenin pathway to play a role in increasing tumour growth and promoting transformation in NSCLC cells [126]. Wang et al. recently reported a novel SETDB1-related pathway in NSCLC: SETDB1 promoted AKT K64 methylation in response to growth factor stimulation, thus promoting methylation-mediated AKT phosphorylation to activate AKT, which promoted NSCLC tumorigenesis [13].

#### 3.1.1. Amplification and Overexpression of SETDB1 in Lung Cancer

Lung cancer is associated with both genetic and epigenetic alterations [2]. In many primary lung tumour cases, amplification of *SETDB1* led to increased levels of *SETDB1* mRNA and protein [11,12,13,14,116,127]. For instance, SETDB1 was confirmed to be amplified and/or upregulated in some NSCLC cell lines such as NCI-H1437, NCI-H1395, A549, Calu-1, SK-MES-1, SK-LU-1, SW-900, and PC14 [12,14,127]. Overexpression and amplification of SETDB1 accelerate the development and tumour invasion of lung cancer [11,12] and are also related to poor prognosis of overall survival and shorter disease-free survival in NSCLC patients [13,116,128]. Therefore, SETDB1 was recently confirmed to hold a moderate diagnostic prediction value for NSCLC [2].

Recently, overexpression of *SETDB1* mRNA was revealed in 1140 ADC and SCC patients compared to noncancerous tissues [2]. An additional report found that the *SETDB1* copy number correlated with the SETDB1 expression level in both SCC and ADC and that amplification of the *SETDB1* gene was more frequent in ADC samples than in SCC samples [121]. Furthermore, the overexpression of SETDB1 promotes the growth of lung cancer cells as well as the invasiveness potential of tumourous cells in vitro and in vivo in nude mouse models [12,13,38].

#### 3.1.2. SETDB1 Oncoprotein or Tumour-Suppressor Status in Different Stages

According to a report, there is a trend correlating higher SETDB1 expression with an advanced pathological state in NSCLC patients [11], which coincides with the situation of *SETDB1* amplification in ADC [128]. However, two research groups found strong expression of SETDB1 in the early stages of NSCLC [115,116]. Interestingly, the high level of *SETDB1* mRNA was found to be sustained at all of the carcinogenic stages, independent of sex or age in NSCLC patients [2], indicating that SETDB1 functions in a pro-oncogenic role or a tumour suppressor role during different stages of tumourigenesis, which provides the possibility to utilize SETDB1 as a biomarker for early diagnosis and a potential therapeutic target in NSCLC. We summarize the oncoprotein and tumour suppressive roles of SETDB1 in lung cancer in the following sections.

#### 3.1.3. Overexpression of SETDB1 Promotes Lung Cancer Growth through Regulating SOD1, LINC00476, p53, and FosB

Compared to normal bronchial epithelial cells, superoxide dismutase 1 (SOD1) mRNA and protein levels were recently reported to be dramatically upregulated in NSCLC cell lines and tissues, which promoted cell cycle progression and inhibited apoptosis in NSCLC [129]. However, *miR-409-3p* suppressed SOD1 expression and oncogenic activities by binding to its *SOD1* 3′ UTR directly. Furthermore, bioinformatic analysis showed that SETDB1 was significantly repressed by *miR-409-3p* but correlated with the expression of SOD1, indicating that SETDB1 may facilitate the proliferation and metastasis of NSCLC cells [129].

Recently, SETDB1, Wnt1, and β-catenin were revealed to be upregulated in NSCLC tissues, while *LINC00476* was downregulated. *LINC00476* acts as a tumour suppressor by downregulating SETDB1 to inhibit proliferation, invasion, and migration in NSCLC [37].

Chen et al. found that a *miR-29s*/SETDB1/*TP53* regulatory circuitry exists in NSCLC [14]. It was previously reported that SETDB1 negatively regulated the expression of p53 [11]. Chen et al. confirmed that p53 positively regulated the transcription of *miR-29s* which directly suppressed the expression of *SETDB1* mRNA and protein. In this circuitry, p53 inhibits the expression of SETDB1 by increasing *miR-29s* expression, while *miR-29s* positively regulates p53 expression by directly targeting SETDB1. SETDB1 suppresses *miR-29s* expression by downregulating p53, and *miR-29s* regulates H3K9 methylation by interacting with SETDB1 and p53.

A recent report found that the reverse expression of SETDB1 and FosB may be the potential pathway of cell proliferation and diffusion in lung cancer [38,130]. Doxorubicin treatment promoted the migration and transformation capacity of A549 cells via inhibition of SETDB1 expression and H3K9me3 activity, during which the induction of FosB expression was also discovered. Further study demonstrated that SETDB1 is directly bound to the promoter of the FosB gene, where it exploits its HMTase activity to negatively regulate FosB expression. Targeting MEK/ERK2 signalling with the inhibitor PD98059 blocked the SETDB1-FosB reverse expression induced by doxorubicin treatment, indicating that SETDB1-mediated FosB expression is regulated by the MEK/ERK2 pathway. Therefore, the activation of the MEK/ERK2-SETDB1-FosB pathway may enhance the transformation and migration of lung cancer during anticancer drug therapy [38].

#### 3.1.4. SETDB1 Inhibits Lung Cancer Metastasis by Regulating SMAD2/3 and EMT

As with many other oncogenes, SETDB1 is also regarded as a crucial inhibitor in lung cancer at certain stages [12,13,115,116,117]. SETDB1 expression is dramatically decreased in highly metastatic sublines of the CL1 lung cancer cell line [115]. Further studies found that SETDB1 functionally cooperated with the TGFb-regulated complex SMAD2/3 to inhibit the expression of Annexin A2 (ANXA2), which plays a crucial role in lung cancer metastasis [115]. This observation demonstrated the tumour-suppressive function of SETDB1 by forming the SETDB1-SMAD 2/3 inhibitory complex in metastatic lung cancer cells [115].

Another study demonstrated that SETDB1 KO by CRISPR/CAS9 significantly increased the migration and transformation of the A549 cell line by regulating the expression of E-cadherin, ꞵ-catenin, STAT3, and AKT, which also indicates the tumour suppressive role of SETDB1 [131]. SETDB1 enzyme deficiency was reported to be unable to inhibit the migration ability of lung cancer cells, indicating that the antitumour function of SETDB1 in lung cancer cells is dependent on its histone H3K9 methylase activity [115].

Epithelial–mesenchymal transition (EMT) plays a pivotal role in the metastasis of NSCLC cells [132]. Yong-Kook Kang et al. showed that an inverse relationship exists between SETDB1 and EMT levels in SCC and ADC [121], which disagrees with the notion that SETDB1 overexpression promotes metastasis of liver [133] and breast cancers [134,135]. Overexpressed SETDB1 also leads to underrepresented genes related to EMT collection [121].

In addition, overexpressed SETDB1 also regulates the transcription of genes related to the host innate immune response, apoptosis, tumour growth and autoimmunity to inhibit growth and autoimmunity in ADCs and SCCs [121].

### 3.2. Malignant Mesothelioma

With a highly unfavorable prognosis, malignant mesothelioma is a rare, incurable, aggressive and highly lethal malignancy originating from the lining layers of the viscera such as the pleura, peritoneum or pericardium [17,136,137,138,139,140]. Mesothelioma is divided into three histological subtypes: epithelioid, sarcomatoid, and biphasic [15]. The pathogenesis of mesothelioma is multifactorial but not fully understood, and the most common inducement is asbestos fibre exposure and inhalation [136,139]. Germline mutations of tumour-suppressing genes have also been confirmed as the main inducements of mesothelioma tumourigenesis [141].

To date, there have been no effective curative strategies for mesothelioma treatment [142]. Although surgery is effective in patients with early, limited and resectable epithelioid mesothelioma [143,144], chemotherapy based on a platinum compound (cisplatin plus antifolate pemetrexed) introduced in 2003 with modest effects remains the only approved first-line regimen for unresectable mesothelioma [145,146,147,148,149]. Immune checkpoint inhibitors such as tremelimumab, pembrolizumab, and nivolumab have shown impressive disease control rates and prolonged disease stability when used as first-, second- or third-line treatments [142,150,151,152,153]. It is noteworthy that the combined immunotherapy of CTLA4 inhibitor nivolumab plus PD-L1 inhibitor ipilimumab has been approved as the first-line therapy for mesothelioma, which presents remarkable extended overall survival compared with chemotherapy [154]. However, immunotherapy does not always represent favourable effects in different subtypes of mesothelioma, and the responses do not always translate into delayed progression or improved survival [149]. Therefore, further studies and trials are needed to determine when it is suitable to apply immunotherapy. Targeted therapies demonstrated a mild response in mesothelioma due to high heterogeneity [155,156,157]. Further biology and occurrence mechanisms of mesothelioma should be studied to explore novel therapeutic targets and effective therapies.

Comprehensive genomic studies have shown a rarity of oncogenic driver mutations in mesothelioma [15,16], while homozygous deletion (HD) and point mutation in tumour suppressor genes (TSGs) are commonly found [148]. Molecular genetic analysis of mesothelioma revealed frequent copy number loss and recurrent somatic mutations in TSGs such as *BRCA1-associated protein 1* (*BAP1*, 60% of the cases), *neurofibromin 2* (*NF2*, 75% of the cases), and *cyclin-dependent kinase inhibitor 2A* (*CDKN2A*, 60% of the cases) [16]. As the most frequent mutations/deletions, the collaborative inactivation of *BAP1*, *NF2*, and *CDKN2A* could result in rapid and aggressive mesothelioma, while homozygous mutation rarely results in carcinogenesis [158,159]. The inactivation of BAP1 also enhanced polycomb repression and enhancer of zeste 2 polycomb repressive complex 2 subunit (Ezh2)-mediated H3K27me3 toward promoter sites [160]. The tumour-suppressing functions of *BAP1*, *KAP1*, *TP53*, *LATS2*, *SETD2*, *USP49* and *PRMT6*, some of which are the genes participating in histone modification and regulation, were also lost in mesothelioma [15,16,161,162,163,164,165,166,167], indicating that the inactivation of histone-modifying genes plays a crucial role in the occurrence and metastasis of mesothelioma [161]. Therefore, demonstration and investigation of genetic alterations, including oncogenic genes [15,162,163,168] and novel tumour suppressors, are critical for further development of diagnosis, prognosis, and personalized therapeutic modalities in mesothelioma.

A study sequencing the transcriptomes and exomes of mesothelioma tissues of 216 patients found 2529 protein-altering somatic mutations in total, among which *SETDB1* was significantly mutated [15]. A 6% mutation frequency of *SETDB1* in 110 patients enrolled in ramucirumab studies was also recently discovered [147]. Moreover, a study of germline mutations in 101 mesothelioma patients suggested that germline missense variants in *SETD2* and *SETDB1* may be associated with predisposition for mesothelioma [17], and frequent inactivating mutations of SETDB1 were found in 78 primary mesothelioma tissues from 69 patients (mutation frequency: 10%, 7/69), indicating the important role of SETDB1 in mesothelioma [18].

In a recent study, four of five near-haploidization mesotheliomas harbouring *TP53* mutations contained point mutations or homozygous deletions of *SETDB1* [16]. The near-haploidization subtype of mesothelioma was also characterized as genome-wide loss of heterozygosity and near-haploid karyotype. This result suggested that early *TP53* mutations allowed chromosome loss to a near-haploid state, followed by genome reduplication and SETDB1 inactivation. A strong correlation between comutations of *TP53* and *SETDB1* and widespread loss of heterozygosity (LOH) was reported [169,170], which was consistent with genome-wide allelic loss or heterozygosity loss [17]. These novel findings suggest that the function of SETDB1 may be correlated with the expression of the tumour suppressor p53 in mesothelioma.

These results reflect that the histone-modifying gene *SETDB1* plays a pivotal role in mesothelioma, and the known mutations of SETDB1 and their deleterious effects are shown in Table 2. Therefore, further studies should be performed to reveal the function and mechanisms of SETDB1. A clinical trial for the drug mithramycin that targets SETDB1 is underway for both lung cancer and mesothelioma (https://clinicaltrials.gov/, accessed on 9 January 2021). To date, there are no studies demonstrating the correlation between SETDB1 and the three most frequent mutated genes (*BAP1*, *NF2*, and *CDKN2A*) in mesothelioma. Further study of the possible interaction and regulatory mechanisms between SETDB1 and these commonly mutated genes during the occurrence and development of mesothelioma is an exciting opportunity.

After analysing the status and potential functions of SETDB1 in lung cancer and mesothelioma, the status of SETDB1 in both primary lung cancer and mesothelioma tissues was recognized as shown in Table 3.

Amplified and overexpressed expression of SETDB1 generally exists in lung cancer [2,11,12,13,14], while inactivation of SETDB1 was discovered in mesothelioma [15,16,17,18,19], indicating that SETDB1 plays disparate roles in different cancer types. SETDB1 may also exist in different statuses during the pathologic process, which demonstrates high heterogeneity [11,115,116,128]. Therefore, SETDB1 may have prospects in diagnosing and developing targeted therapies for both lung cancer and mesothelioma, which should be assessed in additional in-depth mechanistic studies.

### 3.3. Brief Summary of the Regulation Mechanisms of SETDB1 in Cancers with a Focus on Lung Caner

As shown in Figure 2A, SETDB1 is activated or inhibited in various cancers and plays a key role as oncoprotein or tumour suppressor [2,12,13,113,114,115,116,117]. As an oncoprotein, overexpression of SETDB1 is widely found in pancreatic ductal adenocarcinomas [109,110], colorectal cancer [14,21,110,111], melanomas [112], Hela cell line of cervical cancer [1], lung cancer [21] and breast cancer [72]. In these cancers, SETDB1 may suppress the promoter region of tumour-suppressive genes by exerting excessive H3K9 trimethylation. However, the downregulation of SETDB1 in metastatic lung cancer [12,13,115,116,117], less aggressive osteosarcomas [21] and AML [68,69,74,118,119,120] indicates SETDB1 is acting as a tumour suppressor. The true function of SETDB1 in mesothelioma is still to be illustrated [16,17]. In lung cancer, SETDB1 was found to be both upregulated and downregulated, indicating that it holds dual functions in specific cancer types [11,12,13,14,115,116,117,127]. SETDB1 expression was revealed to correlate with different stages in different NSCLC patients [2,11,115,116,128]. On the contrary, a series of studies revealed that only mutation and inactivation of SETDB1 exist in mesothelioma, indicating that SETDB1 may play a tumour-supressing role [15,16,17,18,19].

We herein summarize the SETDB1 mechanisms previously discussed in lung cancer (Figure 2B). The expression of SETDB1 in lung cancer depends on *SETDB1* state and is regulated by interactive factors. Self-mutation may lead to inactivation of SETDB1. SETDB1 is significantly repressed by *miR-409-3p*, while correlating with the expression of SOD1 [129]. SETDB1 is downregulated by tumour suppressor *LINC00476* [37]. The *miR-29s*/SETDB1/*TP53* regulatory circuitry existing in NSCLC signifies that *miR-29s* directly downregulate SETDB1 while p53 inhibits the expression of SETDB1 by increasing the expression of *miR-29s* [14]. In an overexpressed or amplified state or when induced by growth factor, the expression of SETDB1 is upregulated (Figure 2B). 

Overexpressed SETDB1 positively stimulates the WNT-β-catenin pathway to promote the growth and transformation of NSCLC cells [126]. In addition, SETDB1 promotes AKT K64 methylation and methylation-mediated AKT phosphorylation, through which it activates AKT and promotes NSCLC tumorigenesis [13]. The correlation between SOD1 and SETDB1 in NSCLC facilitates the proliferation, invasion and metastasis of NSCLC cells [129]. Moreover, the reverse expression of SETDB1 and FosB mediated by MEK/ERK2 in anticancer drug therapy may be the potential pathway of cell proliferation, migration, transformation capacity, and diffusion in lung cancer [38,130]. As a tumour suppressor, SETDB1 cooperates with SMAD2/3 complex to form the SETDB1-SMAD 2/3 inhibitory complex in metastatic lung cancer cells [115]. The inverse relationship between SETDB1 and EMT levels in SCC and ADC was revealed, in which SETDB1 represses the migration and transformation of lung cancer, as demonstrated in (Figure 2B) [121].

## 4. Future Prospects

Different expression levels of SETDB1 in different cancer types and stages indicate that SETDB1 may be a diagnostic biomarker in certain cancers, such as lung cancer and mesothelioma. In the context of lung cancer, amplification or overexpression of SETDB1 plays an oncogenic role, providing an opportunity to target SETDB1 with inhibitors in clinical treatment. SAM, which is the methyl group donor, is involved in the mechanism of SETDB1 mediated methylation. Thus, small molecules could compete with SAM to inhibit SETDB1 activity. With this in mind, development strategies of SETDB1 competitive inhibitors may be available. In addition, SETDB1 exploits its gene-silencing function through protein complexing, indicating regulation of SETDB1-associated molecular signatures could facilitate the development of targeted therapies for particular types of lung cancer. Moreover, in cancers, transposable elements and immune clusters were discovered to be silenced by SETDB1-dependent H3K9 methylation, which identifies SETDB1 as a negative regulator for tumour-intrinsic immunity. This discovery indicates that application of SETDB1 as a candidate target for immunotherapy is a promising direction for further research.

Frequent SETDB1 inactivation mutations indicate the diagnostic or treatment potential for SETDB1 in mesothelioma. Gene therapy targeting *SETDB1* mediated by oncolytic viruses or AAV viruses may have potential. In addition, there may be specific regulatory mechanisms between SETDB1 and p53, which should be further identified.

Above all, SETDB1 shows promise as a target in multiple targeted and more effective personalized therapies, but further studies, including experiments in animal models, are still needed to confirm whether it is feasible and suitable for use in clinical therapies.

## Figures and Tables

**Figure 1 ijms-22-07416-f001:**
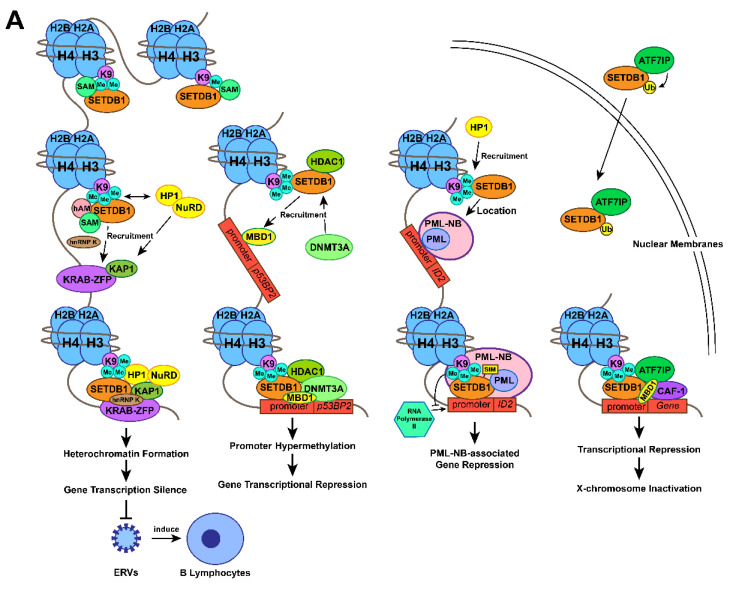
Action mechanisms and the subsequent functions of SETDB1 with its cofactors. (**A**) Together with H3K9 trimethylation on specific gene locus, the formation of SETDB1-KRAB-KAP1, SETDB1-DNMT3A, SETDB1-PML, SETDB1-ATF7IP-MBD1 complexes play crucial roles in the processes of histone methylation, chromatin remodelling and transcriptional suppression, which leads to corresponding physiological functions, including ERVs silencing, XCI and tumour-suppressive genes repression. (**B**) SETDB1 exerts different roles during the preimplantation stage due to its varying expression patterns. It maintains the pluripotency of mESCs by repressing differentiation-associated and trophoblastic genes while facilitating self-renewal by positively regulating the genes associated with iPSC reprogramming and self-renewal.

**Figure 2 ijms-22-07416-f002:**
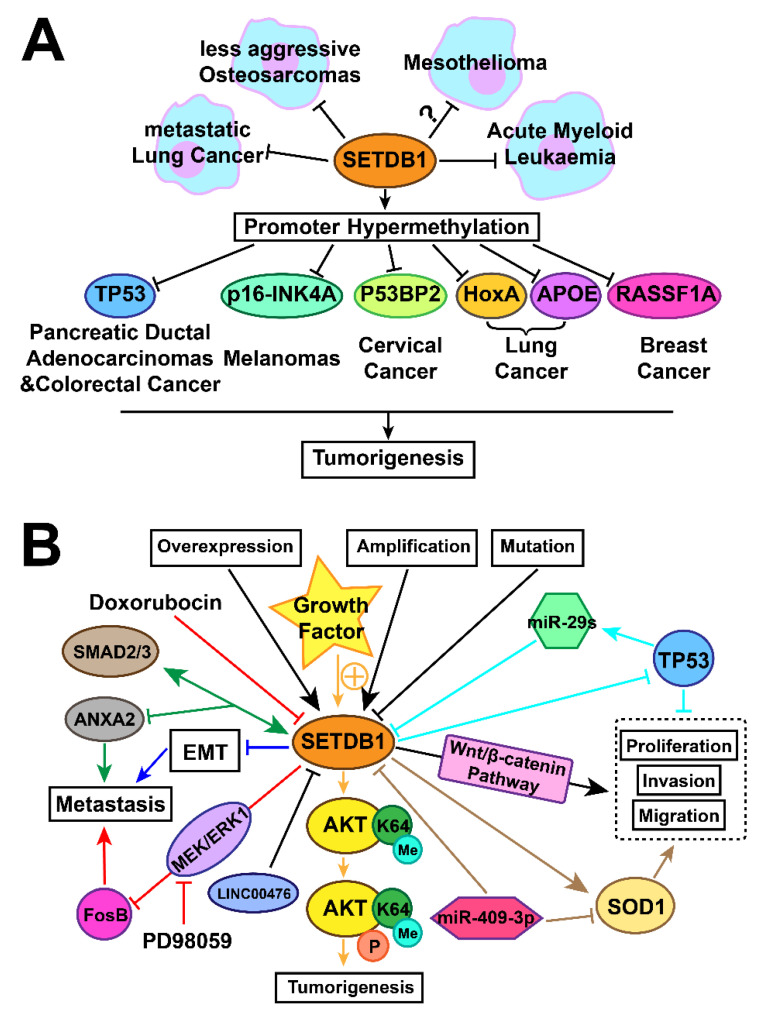
Functions and regulatory mechanisms of SETDB1 in different cancers. (**A**) The expression of SETDB1 differs in various cancers, indicating that SETDB1 may act as oncoprotein or tumour suppressor. (**B**) In lung cancer, SETDB1 was downregulated by self-mutation, *miR-29s*/SETDB1/*TP53* circuitry, *LINC00476*, and *miR-409-3p*, while SETDB1 was upregulated under overexpression/amplification or induced by growth factor. SETDB1 facilitates the development of lung cancer by positively regulating SOD1, AKT methylation and the WNT/β-catenin pathway while negatively regulating the expression of FosB. Additionally, SETDB1 connects with SMAD2/3 and downregulates EMT biomarkers to repress the development of lung cancer.

**Table 1 ijms-22-07416-t001:** Common histone methyltransferases and their related cancers.

Histone Methyltransferase	Modification Type	Links to Cancer	References
MLL2	H3K4me1	Loss and deleterious mutations in NSCLC; Haematopoietic malignancies	[23,24]
SET1B	H3K4me3	Regulation of transcription to maintain stem cell identity	[3,24,25,26]
ET1A	H3K4me1/me2/me3		[26]
ASH1	H3K4me1/me2/me3		[26]
G9a	H3K9me1/me2	Overexpression of G9a promotes metastasis of lung cancer cells	[23,26]
SUV39H2	H3K9me2/me3	Promotion of the proliferation and metastasis in intestinal cancer cells; Inhibition of gastric cancer and lung cancer	[8,26]
SETDB1	H3K9me2/me3	Promotion of liver cancer; CNS diseases	[26,27,28]
SETDB2	H3K9me1/me2/me3	Inhibition of gastric cancer and haematologic malignancies	[8,26]
CLL8	H3K9me1/me2/me3	Obesity and fatty liver	[26,27]
EZH2	H3K27me1/me2/me3	Overexpression in lung cancer; Malignant tumours of the haematopoietic system and hepatocellular carcinoma	[3,23,27,29]
SMYD2	H3K36me1/me2/me3	Proliferation of lung cancer through ALK activation; Contribution to NSCLC cell growth	[23]
SETD2	H3K36me1/me2/me3	Inhibition of lung cancer; Harmful mutations in primary NSCLC	[23]
WHSC1LI	H3K36me1/me2/me3	Overexpression in lung cancer	[23]
SET3	H3K36me1/me2/me3		[26]
NSD1/2/3	H3K36me1/me2/me3		[26]
DOT1L	H3K79me1/me2/me3	Promotion of NSCLC cell growth	[3,23]
SUV4-20H1/2	H4K20me3	Decreased H4K20me3 in tumour progression	[23]
NSD1	H4K20me1/me2/me3		[26]
PRMTs	Arginine on H3 and H4	Overexpression in TKI resistant NSCLC promotes NSCLC growth	[23]

**Table 2 ijms-22-07416-t002:** Mutations of SETDB1 and deleterious effects.

Mutation Modalities	Nucleotide Alterations	Protein Alterations	Mutation Location	Effects	References
Artificial single-amino acid substitution (Base substitution)		H1224K	C-terminal (SET domain)	Impair histone H3 methylase activity; accelerate melanoma	[9,108]
		C1226A	C-terminal (SET domain)	Impair H3 methylase activity; accelerate melanoma	[9,108]
		C1279Y	C-terminal (SET domain)	Impair H3-methylase activity	[9]
Spontaneous mutations in MPM patients (Base substitution)	2606G>A Missense	G869E	C-terminal (Bifurcated SET)	(Damaging)	[17]
	2840C>G Missense	S947C	C-terminal (Bifurcated SET)	Unknown	[17]
	2732G>T Missense	C911F	C-terminal (Bifurcated SET)	Unknown	[17]
	747T>A Nonsense	Y249X	N-terminal	Loss of function; Potential role in MPM development	[17]
Spontaneous mutations in MPM patients (Deletion mutations)	677_693del17 Frameshift (duplicate)	P226RfsX4	N-terminal	Loss of function (duplicate mutation)	[17]
		P226RfsX4	N-terminal		[17]
	3747_3749del In-frame deletion	F1250del	C-terminal (Post-SET)	Unknown	[17]
	2020delA In-frame deletion	K674SFSX73	C-terminal (Between Pre-SET and MBD domian)	Loss of function	[163]
	395_399del5 Frameshift	V132FS	N-terminal	Loss of function; premature stop codon in MPM development	ACCMESO1 mesothelioma cell line (Cancer Cell Line Encyclopedia (CCLE) database)

**Table 3 ijms-22-07416-t003:** SETDB1 status in primary lung cancer and mesothelioma.

Study Design and Subjects	Conclusion	Reference
Primary tumours of lung cancer patients at different grades (*n* = 192) and adjacent normal tissues (*n* = 16)	SETDB1 overexpression in lung cancer (especially in early ones)	[115]
Primary NSCLC at Stage I (*n* = 64) and adjacent normal tissues	Poor prognostic marker of high *SETDB1* mRNA in early NSCLC	[116]
Eight microarrays from GEO and Expression Atlas Databases; Primary NSCLC (*n* = 60) and their paired adjacent normal tissues (*n* = 60)	SETDB1 overexpression in NSCLC (especially in advanced ones)	[11]
Lung cancer tissues (*n* = 387) and normal bronchial epithelium cells (*n* = 106)	SETDB1 overexpression in NSCLC (especially in advanced ones)	[11]
Primary ADC (*n* = 164) and SCC (*n* = 99) tissues	High-level *SETDB1* gene amplification in ADC tissues (especially in advanced ones); poor survival marker of *SETDB1* gene amplification in ADC; low-level *SETDB1* gene amplification in SCC tissues	[128]
Primary ADC (*n* = 20), SCC (*n* = 20), SCLC (*n* = 19) tissues	Gene amplification and high protein level of SETDB1 in NSCLC and SCLC	[12]
TCGA ADC dataset	Gene amplification and high protein level of SETDB1 in NSCLC	[11]
NSCLC (*n* = 1140) and controls (*n* = 952)	High expression of SETDB1 in NSCLC	[2]
12 microarray datasets of NSCLC patients, including current smoker (*n* = 297), former smoker (*n* = 547) and non-smoker (*n* = 220)	Higher *SETDB1* mRNA expression in patients with smoking history	[2]
NSCLC tissues in cBioPortal and Oncomine database (*n* = 1926)	High *SETDB1* mRNA expression in NSCLC (a poor prognostic marker)	[13]
Primary NSCLC tissues (*n* = 9) and paired adjacent normal tissues (*n* = 9)	SETDB1 overexpression in NSCLC	[13]
Primary NSCLC tissues (*n* = 156)	High SETDB1 expression in NSCLC	[13]
34 microarray datasets of ADC and SCC patients	Association between SETDB1 expression and *TP53* mutations in NSCLC	[2]
Primary NSCLC tissues (*n* = 30) in Stage III and IV and paired adjacent normal tissues (*n* = 30)	*SETDB1* mRNA upregulation in primary NSCLC tissues; negative correlation between *SETDB1* and *TP53* mRNA levels	[14]
Primary MPM tissues samples with no prior systemic therapy (*n* = 74) from The Cancer Genome Atlas (TCGA); Japanese International Cancer Genome Consortium (ICGC) MPM cohort (*n* = 80)	Inactivation of SETDB1 in MPM; a novel genomic subtype with *TP53* and *SETDB1* mutations and extensive loss of heterozygosity	[16]
Transcriptomes (*n* = 211), whole exomes (*n* = 99), whole-genome (*n* = 1) and targeted exomes (*n* = 103) from human primary MPM tumour tissues (*n* = 217)	Recurrent and significant *SETDB1* mutations in MPM	[15]
MPM patients enrolled in the Ramucirumab Mesothelioma clinical trial (RAMES) (*n* = 110)	*SETDB1* mutation in MPM	[15]
78 MPM tissues from MPM patients (*n* = 69)	Frequent *SETDB1* mutations in MPM	[17]
Genetic variation of Japanese with mesothelioma based on the exome sequencing (*n* = 1208) and on genotyping data of common variations (*n* = 3248) from the Human Genetic Variation Database (HGVD)	Germline variants and rare missense variants of *SETDB1* in mesothelioma	[18]
A multicentric retrospective case-control cohort of surgically resected MPMs (*n* = 69)	*SETDB1* mutations in epithelioid and biphasic MPM	[19]

## Data Availability

Not applicable.

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
