# Peer review of "The Updating of Biological Functions of Methyltransferase SETDB1 and Its Relevance in Lung Cancer and Mesothelioma"

_ijms, 2021, doi:10.3390/ijms22147416_

Round 1
Reviewer 1 Report
The authors reviewed the biological function of SET domain bifurcated 1 (SETDB1) depending on the different binding partners and different mutation status of SETDB1 between lung cancer and mesothelioma. This is an up-to-date work, and it brings some inspiring points like the involvement and control of the endogenous retrovirus effect on the escape of immune checkpoints and the dual role for SETDB1 as either oncoprotein or tumor suppressor depending on tumor stage and other factors.
However, there are concerns regarding the appropriateness of the references. For example:
1) in Table 1, SETDB1 raw refs 28-30 are related to mesothelioma development, while these papers don't seem to report any data on mesothelioma. Either a ref on mesothelioma (eg #20) needs to be added, or the mention of mesothelioma must be removed.
2) lines 55-57, ref #39 is specifically on hepatocellular carcinoma (HCC) and not all cancers, as stated in the sentence. A new, more appropriate reference is needed here.
3) lines 411-416, ref #19 is on somatic mutations, while the paragraph describes the work of Yoshie in germline mutations of a 101 patients cohort. Thus the reference of Yoshie et al (ref #20) must be quoted instead, as appropriately done in Table 3. There seems a confusion among the authors. Please double-check carefully all the references that have been quoted to make sure they are accurate.
Others:
a) for sake of clarity and accuracy, please move the references [12–14,17,19,20,40,41] at line 61, right after the sentence ' in lung cancer and malignant mesothelioma' at line 59.
b) at the beginning of line 119, the correct sentence must be 'embryonic stem cells (mESCs)'. Please ask to change.
c) title at line 371, remove the 'r' at end of 'mesothelioma'. Moreover, the related pathways and mechanisms are quite complicated. Therefore, if the authors can draw figure(s) to help explaining the interaction of various molecules and proteins, it will help the readers understand the mechanisms better.
Author Response
We are very appreciative your constructive criticisms, and we believe the revised manuscript is substantially improved. We have responded to each of the criticisms, as detailed here:
1) We have checked refs 28-30 and indeed did not mention the relevant data of mesothelioma. Therefore, the contents related to mesothelioma in Table 1 have been deleted. In addition, we rechecked the relevant reference and revised the contents related to SMYD2.
2) We were negligent in quoting references previously, so ref #39 was replaced by ref #23 at line 57. Thank you for your correction.
3) In our description of their experiment, ref #19 was cited by mistake and we changed it to reference 20 at line 514.
Others:
a) We have moved the references [12–14,17,19,20,40,41] at line 61, right after the sentence ' in lung cancer and malignant mesothelioma' at line 59.
b) We changed sentence to 'embryonic stem cells (mESCs)' at the beginning of line 120.
c) We removed the 'r' at end of 'mesothelioma' at line 462.
Moreover, to clearly demonstrate the mechanisms of SETDB1, including the interactions of SETDB1 and parterns, and the related signaling molecules, we have added Figure 1 (page 8-9). We also added Figure 2 to demonstrate the functions and regulatory mechanisms of SETDB1 in different cancers(page 16).
We would be delighted to respond to any additional criticisms that might arise in re-review of the manuscript, and – again – we are most grateful to have this opportunity to resubmit our manuscript.
With best wishes and many thanks.
Yours sincerely,
Wen-Bin Ou, PhD
Reviewer 2 Report
The review deals with a very interesting subject that is the different roles of a specific methyltransferase and of its relevance in lung cancer and mesothelioma. Undoubtely it deserves to be published, but, before this, many comments should be addressed.
General comments:
The English language must be exstensively revised, for example :
Page 3 line 76: SETDB1 dimethylates H3K9 alone, ……..
Page 12, point 3. I think that the text should be: …SAM, which is the methyl group donor, is involved in the mechanism of SETDB1 mediated methylation
Page 3 line 77: in all the cases in which it is presented the new name of a cofactor or other proteins, such as: human homologue of murine ATFa-associated modulator (hAM) , the name should be in italics, to distiguish the text from the name. Another example in line 144 of page 4.
The description of all the very complex roles plaid by SETDB1 occupies the main part of the review, while only a minor part deals with its role in cancer. I suggest, for making more immediate and understandable the text of the first part, to employ the help of graphic tools if and when possible.
Minor comments.
Title: in my opinion the title is too ambitious. The therapeutic relevance of SETDB1 is no more that a hypothetic possibility, therefore I suggest to change it as follows: The updating of biological functions of methyltransferase SETDB1 and its relevance in lung cancer and mesothelioma.
Table 1: first line, loss and deleterious mutations in NSCLC
Page 4, line 126: which brain cells are affected? Which type of cells are iMEF?
Page 6, Line 227: as mentioned in ??
Major comments.
In my opinion the review should include not only a list of effects, but also if possible, in each paragraph, a conclusive sentence, which is not always reported in the present work. This may help to better understand the complex role and relevance of SETDB1.
For example: a conclusive sentence in the 2.6 paragraph may be useful.
page 5, paragraph 2.5. The role plaid by SETDB1 in the X inactivation is undoubtely important, but it cannot be presented as the only one. A lot of papers showed that a lncRNA is involved. Therefore this should be at least mentioned. Otherwise the reader could believe that SETDB1 may represent the only mechanism involved in X inactivation.
Page 6, line 224: please substitute with….to be necessary in meiosis and early oogenesis, without it, a decrease in the number of mature eggs results.
Page 6, line 249. At this point, if possible, it could be important for the reader to see summarized in a graphic scheme the roles of SETDB1 described before.
Page 6, lines 256-260. These sentences should be better explained
Page 6, line 260. The silencing of tumor suppressive genes requires some examples.
Page 6, lines 261-262. Again examples are required.
Page 7, line 3.1.1 paragraph. The authors don’t address specifically the mechanism of activation of SETDB1. Is it only dependent on SETDB1 expression? Or is it also dependent on other regulation (as also mentioned by the authors) involving SAM (and its availability)? The same comment is also valid for SETDB1 inactivation/downregulation due to either the relationship with p53 or the interaction with LINC0047. These circuits and mutual interactions are complex, and again I suggest to use graphic scheme to make them more clear to the reader. Besides the requirement to make more clear the subject, I think that the activation/overexpression and the inhibition/downregulation of SETDB1 would deserve a specific paragraph.
Page 7, lines 321-326. These sentences should be better explained and simplified.
Page 12, point 3. I think that the text should be: …SAM, which is the methyl group donor, is involved in the mechanism of SETDB1 mediated methylation
Page 12, line 465. I suggest to add the following sentence,……further studies, including experiments in animal models,……
Author Response
We are very appreciative your constructive criticisms, and we believe the revised manuscript is substantially improved. We have responded to each of the criticisms, as detailed here:
General comments:
1) English language and style
Page 4, line 76: We double checked English language and style in the whole manuscript. The whole paper was revised by a native American expert (Ms. Isabella Klooster). We substituted "SETDB1 individually exerts dimethylation on H3K9..." for the original expression.
Page 17, line 619-621: According to your suggestion. We reorganized the text as "SAM, which is the methyl group donor, is involved in the mechanism of SETDB1 mediated methylation." At the same time, Section 4 has be rephrased to make it easier for readers to understand.
Page 4 line 74, page 4 line 77, page 4 line 85-86, page 4 line 88, page 4 line 93, page 5 line 123, page 5 line 148, page 6 line 194, page 6 line 198, page 7 line 259-260, page 7 line 266-267, page 12 line 492-493, page 12 line 497: We changed all the new names of cofactors or other proteins in italics in the whole passage.
2) Graphics application
To clearly demonstrate the action mechanisms of SETDB1, including the interactions between SETDB1 and partners, and the related signaling molecules, we have added Figure 1 (page 8-9). To visually illustrate the functions and regulatory mechanisms of SETDB1 in different cancers, we added Figure 2 (page 16).
Minor comments:
1) We agree with your opinion on the title, so, we alter the title into "The updating of biological functions of methyltransferase SETDB1 and its relevance in lung cancer and mesothelioma" as you suggested.
2) We corrected the spelling mistake into "loss and deleterious mutations in NSCLC" in Table 1.
3) Page 5, line 127-128: We further referred to the studies, and confirmed that neural progenitor cells (NPCs ) was mentioned. iMEF refers to immortalized mouse embryonic fibroblasts. We made corresponding modifications in our passage.
4) Page 7, Line 250-251: Due to the problem in format, it failed to show "as mention in 2.7." We made correction.
Major comments:
1) Section 2.2, section 2.3, section 2.4, section 2.5 and section 2.6: We made conclusive sentences respectively at the end of these sections to give clear conclusions.
2) Page 6, line 188-191: We supplemented the correlation between lncRNA and XCI to avoid misunderstanding.
3) Page 7, line 246-248: We substitute the original text with "to be necessary in meiosis and early oogenesis, without it, a decrease in the number of mature eggs results."
4) Page 8: To clearly demonstrate the mechanisms of SETDB1, including the interactions of SETDB1 and partners, and the related signaling molecules, we have added Figure 1.
5) Page 9, line 332-334 : The sentences are better explained as "Under various carcinogenic conditions or regulations of related genes...".
6) Page 9, Line 337-340: We listed some examples to confirm that SETDB1 could promote tumour development by silencing tumour suppressive genes.
7) Page 15-16: In order to demonstrate the mechanisms of inactivation/downregulation of SETDB1, as well as the functions and regulatory mechanisms of SETDB1 in different cancers, we added section 3.3. "Brief summary of the regulation mechanisms of SETDB1 in cancers, especially in lung caner". We also added Figure 2 to offer a clear understanding on the functions and regulatory mechanisms of SETDB1 in different cancers. The mechanisms of inactivation/downregulation of SETDB1 are presented at Page 15, line 567-575.
8) Page 11, line 412-420 : The sentences are better organized and explained as "Chen et al. found that a miR-29s/SETDB1/TP53 regulatory circuitry exists in NSCLC. It was previously reported that SETDB1 negatively regulated the expression of p53. Chen et al. confirmed that p53 positively regulated the transcription of miR-29s which directly suppressed the expression of SETDB1 mRNA and protein. In this circuitry, p53 inhibits the expression of SETDB1 by elevating the expression of miR-29s, while miR-29s positively regulates p53 expression by directly targeting SETDB1. SETDB1 suppresses miR-29s expression by downregulating p53, and miR-29s regulates H3K9 methylation by interacting with SETDB1 and p53".
9) Page 17, line 637: We modified the sentences as following" …further studies, including experiments in animal models,…"
We would be delighted to respond to any additional criticisms that might arise in re-review of the manuscript, and – again – we are most grateful to have this opportunity to resubmit our manuscript.
With best wishes and many thanks.
Yours sincerely,
Wen-Bin Ou, PhD
Reviewer 3 Report
Malignant mesothelioma is described as epitheliod, sarcomatoid or biphasic (please check line 374-375).
What'a about BAP-1 in mesothelioma? correlation with SETDB1?
Section 4 should be rephrased to avoid the points presence.
Please check english to avoid some typoes (e.g. line 371)
Author Response
We are very appreciative your constructive criticisms, and we believe the revised manuscript is substantially improved. We have responded to each of the criticisms, as detailed here:
1) We have corrected the subtype classification of mesothelioma and divided it into three histological subtypes (epitheliod, sarcomatoid or biphasic) at line 465-467.
2) According to your suggestion, we expanded certain background knowledge about the mutations/deletions of BAP1, NF2, and CDKN2A in mesothelioma. We also supplemented the related downstream effects of BAP1 inactivation. Besides, no studies have revealed the correlation between SETDB1 and the three most frequent mutated genes (BAP1, NF2, and CDKN2A) in mesothelioma. We made a statement on it as well, and deem it as an interesting subject. (line 493-498, 529-533)
3) Section 4 has be rephrased to make it easier for readers to understand.
4) We removed the 'r' at end of 'mesothelioma' at line 462. And we double checked English language and style in the whole manuscript.
We would be delighted to respond to any additional criticisms that might arise in re-review of the manuscript, and – again – we are most grateful to have this opportunity to resubmit our manuscript.
With best wishes and many thanks.
Yours sincerely,
Wen-Bin Ou, PhD
Reviewer 4 Report
The authors have submitted a fine review of a difficult subject.
1) In the discussion of mesothelioma therapy with immune checkpoint inhibitors (lines 387-388), the authors need to mention that as of 2021 the combination of ipilimumab and nivolumab (a CTLA4 and PD-L1 inhibitor respectively) has become the standard first-line therapy for mesothelioma (based on the trial of Baas, et al. Lancet. 2021 Jan 30;397(10272):375-386).
2) Line 371, mesothelioma is misspelled as "mesotheliomar"
Author Response
We are very appreciative your constructive criticisms, and we believe the revised manuscript is substantially improved. We have responded to each of the criticisms, as detailed here:
1) We have added that as of 2021 the combination of ipilimumab and nivolumab has become the standard first-line therapy for mesothelioma.The original text is“It is noteworthy that the combined immunotherapy of CTLA4 inhibitor nivolumab plus PD-L1 inhibitor ipilimumab has approved to be the first-line therapy for mesothelioma, which presents remarkable extended overall survival compared with chemotherapy”at line 478-481. And cited reference from Baas, et al. Lancet. 2021 Jan 30;397(10272):375-386.
2) We removed the 'r' at end of 'mesothelioma' at line 462.
We would be delighted to respond to any additional criticisms that might arise in re-review of the manuscript, and – again – we are most grateful to have this opportunity to resubmit our manuscript.
With best wishes and many thanks.
Yours sincerely,
Wen-Bin Ou, PhD
Round 2
Reviewer 2 Report
The authors addressed well all the reviewer's comments.
Reviewer 3 Report
nn